# REVISITING PRIORITIZED EXPERIENCE REPLAY: A VALUE PERSPECTIVE

## ABSTRACT

Reinforcement learning (RL) agents need to learn from past experiences. Prioritized experience replay that weighs experiences by their *surprise* (the magnitude of the temporal-difference error) significantly improves the learning efficiency for RL algorithms. Intuitively, *surprise* quantifies the unexpectedness of an experience to the learning agent. But how *surprise* is related to the importance of experience is not well understood. To address this problem, we derive three value metrics to quantify the importance of experience, which consider the extra reward would be earned by accessing the experience. We theoretically show these value metrics are upper-bounded by *surprise* for Q-learning. Furthermore, we successfully extend our theoretical framework to maximum-entropy RL by deriving the lower and upper bounds of these value metrics for soft Q-learning, which turn out to be the product of *surprise* and "on-policyness" of the experiences. Our framework links two important quantities in RL, *i.e.*, *surprise* and value of experience, and provides a theoretical basis to estimate the value of experience by *surprise*. We empirically show that the bounds hold in practice, and experience replay using the upper bound as priority improves maximum-entropy RL in Atari games.

## 1 INTRODUCTION

Learning from important experiences prevails in nature. In rodent hippocampus, memories with higher importance, such as those associated with rewarding locations or large reward-prediction errors, are replayed more frequently (Michon et al., 2019; Roscow et al., 2019; Salvetti et al., 2014). People who have more frequent replay of high-reward associated memories show better performance in memory tasks (Gruber et al., 2016; Schapiro et al., 2018). A normative theory suggests that prioritized memory access according to the utility of memory explains hippocampal replay across different memory tasks (Mattar & Daw, 2018). As accumulating new experiences is costly, utilizing valuable past experiences is a key for efficient learning (Ólafsdóttir et al., 2018).

Differentiating important experiences from unimportant ones also benefits reinforcement learning (RL) algorithms (Katharopoulos & Fleuret, 2018). Prioritized experience replay (PER) (Schaul et al., 2016) is an experience replay technique built on deep Q-network (DQN) (Mnih et al., 2015), which weighs the importance of samples by their *surprise*, the magnitude of the temporal-difference (TD) error. As a result, experiences with larger *surprise* are sampled more frequently. PER significantly improves the learning efficiency of DQN, and has been adopted (Hessel et al., 2018; Horgan et al., 2018; Kapturowski et al., 2019) and extended (Daley & Amato, 2019; Pan et al., 2018; Schlegel et al., 2019) by various deep RL algorithms. *Surprise* quantifies the unexpectedness of an experience to a learning agent, and biologically corresponds to the signal of reward prediction error in dopamine system (Schultz et al., 1997; Glimcher, 2011), which directly shapes the memory of animal and human (Lisman & Grace, 2005; McNamara et al., 2014). However, how *surprise* is related to the importance of experience in the context of RL is not well understood.

We address this problem from an economic perspective, by linking *surprise* to value of experience in RL. The goal of RL agent is to maximize the expected cumulative reward, which is achieved through learning from experiences. For Q-learning, an update on an experience will lead to a more accurate prediction of the action-value or a better policy, which increases the expected cumulative reward the agent may get. We define the value of experience as the increase in the expected cumulative reward resulted from updating on the experience (Mattar & Daw, 2018). The value of experience quantifies

the importance of experience from first principles: assuming that the agent is economically rational and has full information about the value of experience, it will choose the most valuable experience to update, which will yield the highest utility. As supplements, we derive two more value metrics, which corresponds to the evaluation improvement value and policy improvement value due to update on an experience.

In this work, we mathematically show that these value metrics are upper-bounded by *surprise* for Q-learning. Therefore, *surprise* implicitly tracks the value of experience, and accounts for the importance of experience. We further extend our framework to maximum-entropy RL, which augments the reward with an entropy term to encourage exploration (Haarnoja et al., 2017). We derive the lower and upper bounds of these value metrics for soft Q-learning, which are related to *surprise* and "on-policyness" of the experience. Experiments in Maze and CartPole support our theoretical results for both tabular and function approximation RL methods, showing that the derived bounds hold in practice. Moreover, we also show that experience replay using the upper bound as priority improves maximum-entropy RL (*i.e.*, soft DQN) in Atari games.

## 2 MOTIVATION

### 2.1 Q-LEARNING AND EXPERIENCE REPLAY

We consider a Markov Decision Process (MDP) defined by a tuple $\{\mathcal{S}, \mathcal{A}, \mathcal{P}, \mathcal{R}, \gamma\}$, where $\mathcal{S}$ is a finite set of states, $\mathcal{A}$ is a finite set of actions, $\mathcal{P}$ is the transition function, $\mathcal{R}$ is the reward function, and $\gamma \in [0, 1]$ is the discount factor. A policy $\pi$ of an agent assigns probability $\pi(a|s)$ to each action $a \in \mathcal{A}$ given state $s \in \mathcal{S}$. The goal is to learn an optimal policy that maximizes the expected discounted return starting from time step $t$, $G_t = r_t + \gamma r_{t+1} + \gamma^2 r_{t+2} + ... = \sum_{i=0}^{\infty} \gamma^i r_{t+i}$, where $r_t$ is the reward the agent receives at time step $t$. Value function $v_\pi(s)$ is defined as the expected return starting from state $s$ following policy $\pi$, and Q-function $q_\pi(s, a)$ is the expected return on performing action $a$ in state $s$ and subsequently following policy $\pi$.

According to Q-learning (Watkins & Dayan, 1992), the optimal policy can be learned through policy iteration: performing policy evaluation and policy improvement interactively and iteratively. For each policy evaluation, we update $Q(s, a)$, an estimate of $q_\pi(s, a)$, by

$$Q_{\text{new}}(s, a) = Q_{\text{old}}(s, a) + \alpha \text{TD}(s, a, r, s'),$$

where the TD error $\text{TD}(s, a, r, s') = r + \gamma \max_{a'} Q_{\text{old}}(s', a') - Q_{\text{old}}(s, a)$ and $\alpha$ is the step-size parameter. $Q_{\text{new}}$ and $Q_{\text{old}}$ denote the estimated Q-function before and after the update respectively. And for each policy improvement, we update the policy from $\pi_{\text{old}}$ to $\pi_{\text{new}}$ according to the newly estimated Q-function,

$$\pi_{\text{new}} = \arg \max_a Q_{\text{new}}(s, a).$$

Standard Q-learning only uses each experience once before disregarded, which is sample inefficient and can be improved by *experience replay* technique (Lin, 1992). We denote the experience that the agent collected at time $k$ by a 4-tuple $e_k = \{s_k, a_k, r_k, s'_k\}$. According to *experience replay*, the experience $e_k$ is stored into the replay buffer and can be accessed multiple times during learning.

### 2.2 VALUE METRICS OF EXPERIENCE

To quantify the importance of experience, we derive three value metrics of experience. The utility of update on experience $e_k$ is defined as the value added to the cumulative discounted rewards starting from state $s_k$, after updating on $e_k$. Intuitively, choosing the most valuable experience for update will yield the highest utility to the agent. We denote such utility as the expected value of backup $\text{EVB}(e_k)$ (Mattar & Daw, 2018),

$$\begin{aligned} \text{EVB}(e_k) &= v_{\pi_{\text{new}}}(s_k) - v_{\pi_{\text{old}}}(s_k) \\ &= \sum_a \pi_{\text{new}}(a|s_k) q_{\pi_{\text{new}}}(s_k, a) - \sum_a \pi_{\text{old}}(a|s_k) q_{\pi_{\text{old}}}(s_k, a) \end{aligned} \tag{1}$$

where $\pi_{\text{old}}$, $v_{\pi_{\text{old}}}$ and $q_{\pi_{\text{old}}}$ are respectively the policy, value function and Q-function before the update, and $\pi_{\text{new}}$, $v_{\pi_{\text{new}}}$, and $q_{\pi_{\text{new}}}$ are those after. As the update on experience $e_k$ consists of policy

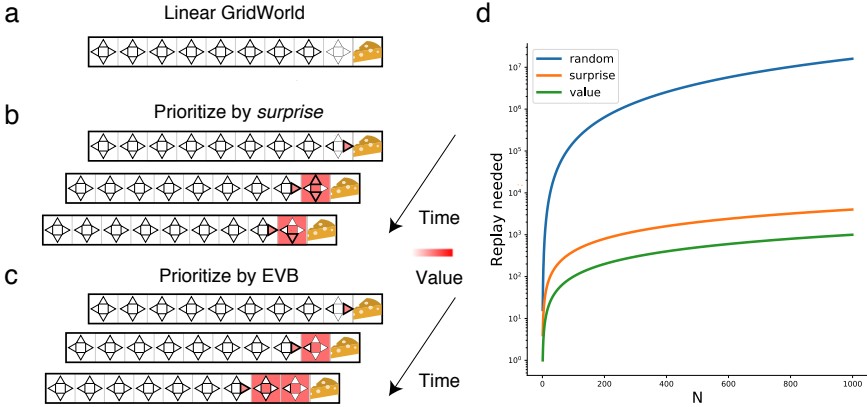

Figure 1: **a.** Illustration of the "Linear GridWorld" example: there are $N$ grids and 4 actions (north, south, east, west). Reward for entering the goal state (cheese) is 1; reward is 0 elsewhere. **b-c.** Examples of prioritized experience replay by *surprise* and value of experience (EVB). The main difference is that EVB only prioritizes the experiences that are associated with the optimal policy; while *surprise* is sensitive to changes in value function and will prioritize non-optimal experiences, such as those associated with north or south. Here squares represent states, triangles represent actions, and experiences associated with the highest priority are highlighted. **d.** Expected number of replays needed to learn the optimal policy, as the number of grids changes: uniform replay (blue), prioritized by *surprise* (orange), and EVB (green).

evaluation and policy improvement, the value of experience can further be separated to evaluation improvement value $\text{EIV}(e_k)$ and policy improvement value $\text{PIV}(e_k)$ by rewriting (1):

$$\text{EVB}(e_k) = \underbrace{\sum_a [\pi_{\text{new}}(a|s_k) - \pi_{\text{old}}(a|s_k)]q_{\pi_{\text{new}}}(s_k, a)}_{\text{PIV}(e_k)} + \underbrace{\sum_a \pi_{\text{old}}(a|s_k)[q_{\pi_{\text{new}}}(s_k, a) - q_{\pi_{\text{old}}}(s_k, a)]}_{\text{EIV}(e_k)}, \quad (2)$$

where $\text{PIV}(e_k)$ measures the value improvements due to the change of the policy, and $\text{EIV}(e_k)$ captures those due to the change of evaluation. Thus, we have three metrics for the value of experience: EVB, PIV and EIV.

### 2.3 VALUE METRICS OF EXPERIENCE IN Q-LEARNING

For Q-learning, we use Q-function to estimate the true action-value function. A backup over an experience $e_k$ consists of policy evaluation with Bellman operator and greedy policy improvement. As the policy improvement is greedy, we can rewrite value metrics of experience to simpler forms. EVB can be written as follows from (1),

$$\text{EVB}(e_k) = \max_a Q_{\text{new}}(s_k, a) - \max_a Q_{\text{old}}(s_k, a). \quad (3)$$

Note that EVB here is different from that in Mattar & Daw (2018): in our case, EVB is derived from Q-learning; while in their case, EVB is derived from Dyna, a model-based RL algorithm (Sutton, 1990). Similarly, from (2), PIV can be written as

$$\text{PIV}(e_k) = \max_a Q_{\text{new}}(s_k, a) - Q_{\text{new}}(s_k, a_{\text{old}}), \quad (4)$$

where $a_{\text{old}} = \arg\max_a Q_{\text{old}}(s_k, a)$, and EIV can be written as

$$\text{EIV}(e_k) = Q_{\text{new}}(s_k, a_{\text{old}}) - Q_{\text{old}}(s_k, a_{\text{old}}). \quad (5)$$

### 2.4 A MOTIVATING EXAMPLE

We illustrate the potential gain of value of experience in a "Linear GridWorld" environment (Figure 1a). This environment contains $N$ linearly-aligned grids and 4 actions (north, south, east, west).

The rewards are rare: 1 for entering the goal state and 0 elsewhere. The solution for this environment is always choosing east.

We use this example to highlight the difference between prioritization strategies. Three agents perform Q-learning updates on the experiences drawn from the same replay buffer, which contains all the ($4N$) experiences and associated rewards. The first agent replays the experiences uniformly at random, while the other two agents invoke the oracle to prioritize the experiences, which greedily select the experience with the highest *surprise* or EVB respectively. In order to learn the optimal policy, agents need to replay the experiences associated with action east in a reverse order.

For the agent with random replay, the expected number of replays required is $4N^2$ (Figure 1d). For the other two agents, prioritization significantly reduces the number of replays required: prioritization with *surprise* requires $4N$ replays, and prioritization with EVB only uses $N$ replays, which is optimal (Figure 1d). The main difference is that EVB only prioritizes the experiences that are associated with the optimal policy (Figure 1c), while the *surprise* is sensitive to changes in the value function and will prioritize non-optimal experiences: for example, the agent may choose the experiences associated with south or north in the second update, which are not optimal but have the same *surprise* as the experience associated with east (Figure 1b). Thus, EVB that directly quantifies the value of experience can serve as an optimal priority.

## 3   Upper Bounds of Value Metrics of Experience in Q-Learning

PER (Schaul et al., 2016) greatly improves the learning efficiency of DQN. However, the underlying rationale is not well understood. Here, we prove that *surprise* is the upper bound of the value metrics in Q-learning.

**Theorem 3.1.** *The three value metrics of experience $e_k$ in Q-learning (|EVB|, |PIV| and |EIV|) are bounded by $\alpha|TD(s_k, a_k, r_k, s'_k)|$, where $\alpha$ is a step-size parameter.*

*Proof.* See Appendix A.1. □

In Theorem 3.1, we prove that |EVB|, |PIV|, and |EIV| are upper-bounded by the *surprise* (scaled by the learning step-size) in Q-learning. As *surprise* intrinsically tracks the evaluation and policy improvements, it can serve as an appropriate importance metric for past experiences. We will further study these relationship in experiments.

## 4   Extension to Maximum-Entropy RL

In this section, we extend our framework to study the relationship between *surprise* and value of experience in maximum-entropy RL, particularly, soft Q-learning.

### 4.1   Soft Q-Learning

Unlike regular RL algorithms, maximum-entropy RL augments the reward with an entropy term: $R = r + \beta\mathcal{H}(\pi(\cdot|s))$, where $\mathcal{H}(\cdot)$ is the entropy, and $\beta$ is an optional temperature parameter that determines the relative importance of entropy and reward. The goal is to maximize the expected cumulative entropy-augmented rewards. Maximum-entropy RL algorithms have advantages at capturing multiple modes of near optimal policies, better exploration, and better transfer between tasks.

Soft Q-learning is an off-policy value-based algorithm built on maximum-entropy RL principles (Haarnoja et al., 2017; Schulman et al., 2017). Different from Q-learning, the target policy of soft Q-learning is stochastic. During policy iteration, Q-function is updated through soft Bellman operator $\Gamma^{\text{soft}}$, and the policy is updated to a maximum-entropy policy:

$$\text{Policy Evaluation: } Q_{\text{new}}^{\text{soft}}(s, a) = [\Gamma^{\text{soft}}Q_{\text{old}}^{\text{soft}}](s, a) = r + \gamma V_{\text{old}}^{\text{soft}}(s')$$

$$\text{Policy Improvement: } \pi_{\text{new}}(a|s) = \text{softmax}_a(\frac{1}{\beta}Q_{\text{new}}^{\text{soft}}(s, a)),$$

where $\text{softmax}_i(x) = \exp(x_i) / \sum_i \exp(x_i)$ is the softmax function, and the soft value function $V_\pi^{\text{soft}}(s)$ is defined as,

$$V_\pi^{\text{soft}}(s) = \mathbb{E}_a\{Q_\pi^{\text{soft}}(s,a) - \log(\pi(a|s))\} = \beta \log \sum_a \exp(\frac{1}{\beta} Q_\pi^{\text{soft}}(s,a)).$$

Similar as in Q-learning, the TD-error in soft Q-learning (soft TD error) is given by:

$$\text{TD}^{\text{soft}}(s,a,r,s') = r + \gamma V_{\text{old}}^{\text{soft}}(s') - Q_{\text{old}}^{\text{soft}}(s,a).$$

## 4.2 VALUE METRICS OF EXPERIENCE IN MAXIMUM-ENTROPY RL

Here, we extend the value metrics of experience to soft Q-learning. Similar as (1), EVB for maximum-entropy RL is defined as,

$$\begin{aligned}
\text{EVB}^{\text{soft}}(e_k) &= v_{\text{new}}^{\text{soft}}(s_k) - v_{\text{old}}^{\text{soft}}(s_k) \\
&= \sum_a \pi_{\text{new}}(a|s_k)\{q_{\text{new}}^{\text{soft}}(s_k,a) - \beta \log(\pi_{\text{new}}(a|s_k))\} \\
&\qquad - \sum_a \pi_{\text{old}}(a|s_k)\{q_{\text{old}}^{\text{soft}}(s_k,a) - \beta \log(\pi_{\text{old}}(a|s_k))\} \qquad (6)
\end{aligned}$$

$\text{EVB}^{\text{soft}}$ can be separated into $\text{PIV}^{\text{soft}}$ and $\text{EIV}^{\text{soft}}$, which respectively quantify the value of policy and evaluation improvement in soft Q-learning,

$$\begin{aligned}
\text{PIV}^{\text{soft}}(e_k) &= \sum_a \pi_{\text{new}}(a|s_k)\{q_{\text{new}}^{\text{soft}}(s_k,a) - \beta \log(\pi_{\text{new}}(a|s_k))\} \\
&\qquad - \sum_a \pi_{\text{old}}(a|s_k)\{q_{\text{new}}^{\text{soft}}(s_k,a) - \beta \log(\pi_{\text{old}}(a|s_k))\} \\
&= \sum_a \{\pi_{\text{new}}(a|s_k) - \pi_{\text{old}}(a|s_k)\}q_{\text{new}}^{\text{soft}}(s_k,a) + \beta(H(\pi_{\text{new}}(\cdot|s)) - H(\pi_{\text{old}}(\cdot|s_k))), \quad (7)
\end{aligned}$$

$$\text{EIV}^{\text{soft}}(e_k) = \sum_a \pi_{\text{old}}(a|s_k)[q_{\text{new}}^{\text{soft}}(s_k,a) - q_{\text{old}}^{\text{soft}}(s_k,a)]. \qquad (8)$$

Value metrics of experience in maximum-entropy RL have similar forms as in regular RL except for the entropy term, because changes in policy leads to changes in the policy entropy and affects the entropy-augmented rewards.

## 4.3 LOWER AND UPPER BOUNDS OF VALUE METRICS OF EXPERIENCE IN SOFT Q-LEARNING

We theoretically derive the lower and upper bounds of the value metrics of experience in soft Q-learning.

**Theorem 4.1.** *The three value metrics of experience $e_k$ in soft Q-learning ($|EVB^{soft}|$, $|PIV^{soft}|$ and $|EIV^{soft}|$) are upper bounded by $\rho_\pi^{max} * |TD^{soft}|$, where $\rho_\pi^{max} = \max\{\pi_{old}(a_k|s_k), \pi_{new}(a_k|s_k)\}$ is a policy related term.*

*Proof.* See Appendix A.2. $\qquad\qquad\square$

**Theorem 4.2.** *For soft Q-learning, $|EVB^{soft}|$ and $|EIV^{soft}|$ (but not $|PIV^{soft}|$) are lower bounded by $\rho_\pi^{min} * |TD^{soft}|$, where $\rho_\pi^{min} = \min\{\pi_{old}(a_k|s_k), \pi_{new}(a_k|s_k)\}$ is a policy related term.*

*Proof.* See Appendix A.3. $\qquad\qquad\square$

The lower and upper bounds in soft Q-learning include a policy term with the *surprise* (the magnitude of the soft TD error). The policy related term $\rho_\pi$ quantifies the "on-policyness" of the experienced action. And the bounds become tighter as the difference between $\pi_{\text{old}}(a_k|s_k)$ and $\pi_{\text{new}}(a_k|s_k)$ becomes smaller. Surprisingly, the coefficient of the entropy term $\beta$ impacts the bound only through

the policy term, which makes it an excellent priority even $\beta$ changes during learning (Haarnoja et al., 2018). As $0 \leq \rho_\pi^{\max} \leq 1$, the value metrics are also upper bounded by *surprise* ($|\text{TD}^{\text{soft}}|$) alone, which is similar as in Q-learning. However, as $\pi(a_k|s_k)$ is usually less than 1, *surprise* is a looser upper bound in soft Q-learning. This supports previous study, which empirically shows that directly applying PER using *surprise* alone in soft Q-learning does not significantly improve the sample efficiency (Wang & Ross, 2019). We will further study these relationship in experiments.

## 5 EXPERIMENTS

Our experiments aim to answer following questions: (*i*) Do the theoretical bounds of value metrics of experience hold true in practice? (*ii*) If the bounds hold, are they tight? (*iii*) Do the bounds derived for maximum-entropy RL help to improve performance? First, we implement tabular versions of Q-learning and soft Q-learning in Maze to verify the bounds in tabular methods. Second, by slightly modifying the definition of the value metrics of experience, we extend our framework to function approximation methods, which is far more powerful than the tabular methods (see Appendix A.4). We implement DQN and soft DQN in CartPole to examine the bounds in function approximation methods. Finally, we implement PER with the theoretical upper bound as priority for soft DQN and evaluate its effectiveness.

Throughout the experiments, the value metrics of experience (EVB, PIV and EIV) for Q-learning are calculated using (3), (4), and (5), and those for soft Q-learning are calculated using (6), (7), and (8). For soft Q-learning, we do not model the actor explicitly: the policy is calculated as the softmax of the soft Q-function (see Section 4.1). The upper bound for value metrics in Q-learning is *surprise* (Theorem 3.1), while the lower and upper bounds for soft Q-learning are calculated according to Theorem 4.1 and 4.2, which include a policy term and *surprise*. The experimental details are described in Appendix A.5 and all the codes are available at: `https://github.com/RLforlife/VER`.

### 5.1 MAZE

The first set of experiments are conducted in a maze environment of a $5 \times 5$ square with walls. The agent needs to reach the goal zone by moving one square in any of the four directions (north, south, east, west) each time. We implement tabular versions of Q-learning and soft Q-learning to solve this problem.

For each algorithm, the value metrics of experience as well as the theoretical bounds are illustrated in Figure 2. As we can see from upper panel, all three value metrics of experience are bounded by the *surprise* for Q-learning. As our theory predicts, the absolute values of EIV are either equal to the *surprise* (if the action of the experience is the best action before update) or 0. For soft Q-learning, the three value metrics of experience are bounded by the theoretical upper bound, and EVB and EIV are bounded by the theoretical lower bound, supporting our theory (Theorem 4.1 and 4.2). There is a large proportion of EVBs lies on the identity line, indicating the bounds are tight. The proportion of non-zero values of experiences is higher in soft Q-learning than in Q-learning, because all value metrics of experiences are affected by the "on-policyness" of the experienced actions. Q-learning learns a deterministic policy that makes most actions of experiences off-policy, while soft Q-learning learns a stochastic policy that results in less sparse values of experiences. In summary, the experimental results in the maze environment support the theoretical bounds of value metrics of experience in Q-learning and soft Q-learning.

### 5.2 CARTPOLE

CartPole is a pendulum with a center of gravity above its pivot point. The goal is to keep the pole balanced by moving the cart forward and backward. We implement DQN and soft DQN (DQN with soft-update) in this environment. For DQN, we replace the Q-network in Mnih et al. (2015) with a two-layer MLP. For soft-DQN, all the settings are the same with DQN, except for two modifications: during policy evaluation, the (soft) Q-network is updated according to the soft TD error; during policy improvement, the policy is updated following a maximum-entropy policy, as the softmax of the Q values (see Section 4.1).

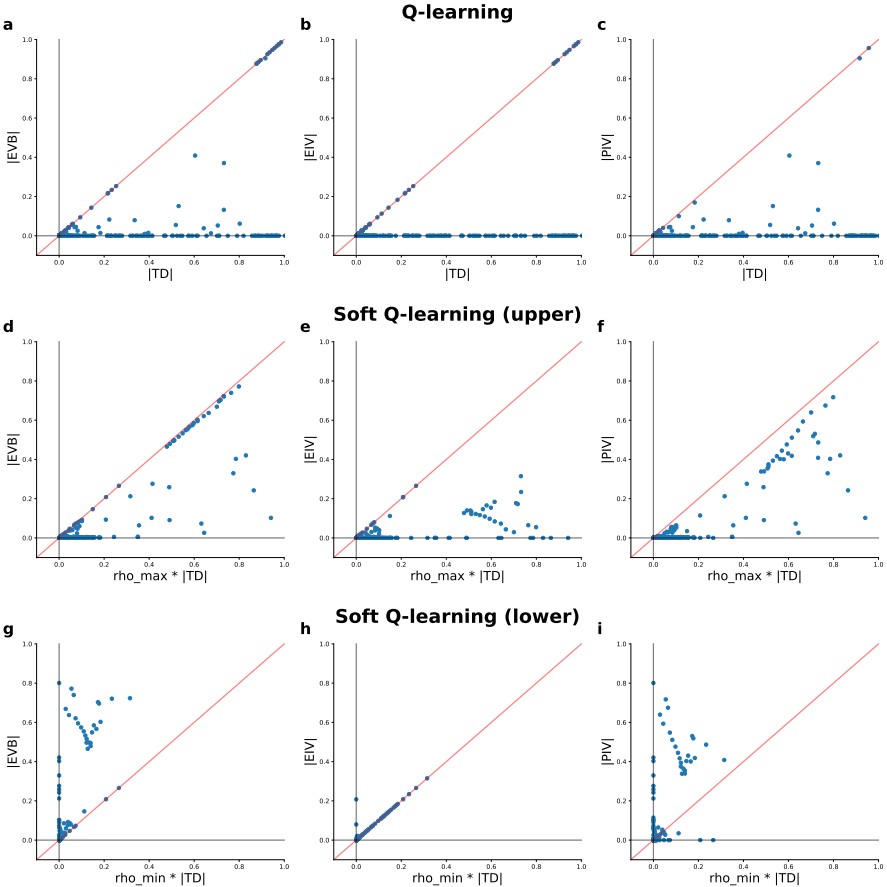

Figure 2: Results of Q-learning and soft Q-learning in Maze. **a-c.** *surprise* (the magnitude of TD error) *v.s.* absolute value of EVB (left), EIV (middle) and PIV (right) in Q-learning. **d-f.** Theoretical upper bound and **g-i.** lower bound *v.s.* absolute value of EVB, EIV and PIV in soft Q-learning. The red line is the identity line.

By slightly modifying the definition of value metrics of experience, we can extend our framework to function approximation methods (see Appendix A.4). The value metrics of experience as well as the theoretical bounds are illustrated in Figure 3. All value metrics of experience in DQN (Figure 3a-c) and soft DQN (Figure 3d-f) are bounded by the theoretical upper bounds. For DQN, absolute EVBs and PIVs are uniformly distributed in the bounded area, while absolute EIVs are equal to the *surprise* or 0. Results are different in soft DQN, where absolute EVBs and EIVs are distributed more closely towards the theoretical upper bounds, suggesting the upper bound in soft Q-learning is tighter. Moreover, (Figure 3g-h) shows the EVBs and EIVs are lower bounded by $\rho_\pi^{\min} * \left|\text{TD}^{\text{soft}}\right|$, while PIVs are not . The experimental results confirm the bounds of value metrics in function approximation methods.

## 5.3 ATARI GAMES

In this set of experiments, we investigate whether the theoretical upper bound of value metrics of experience, which balances the *surprise* and "on-policyness" of the experience (Figure 6), can serve as an appropriate priority for experience replay in soft Q-learning. More specifically, we compare the performance of soft DQN with different prioritization strategies: uniform replay, prioritization with *surprise* or the theoretical upper bound ($\rho_\pi^{\max} * \left|\text{TD}^{\text{soft}}\right|$), which are denoted by soft DQN, PER and VER (valuable experience replay) respectively. This set of experiments consists of nine games from Atari 2600 games, whose goal is learning to play each of the games with the screen pixels as the only input. We closely follow the experimental setting and network architecture outlined by

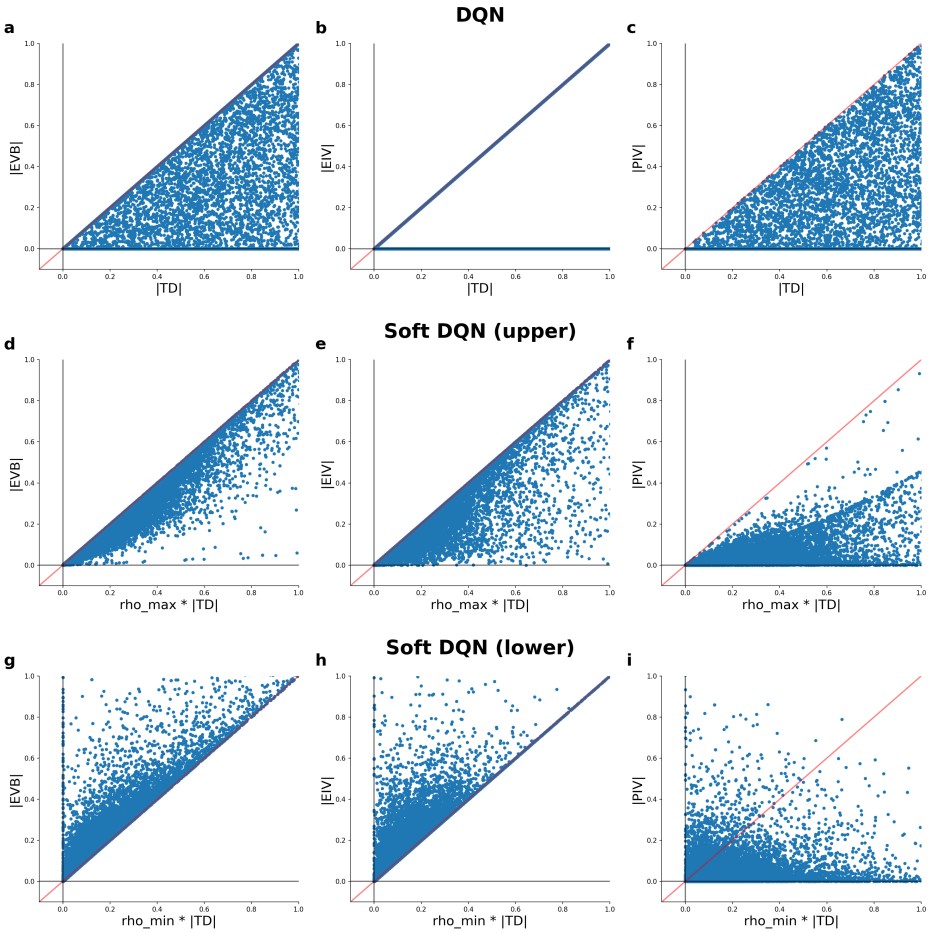

Figure 3: Results of DQN and soft DQN in CartPole. **a-c.** *surprise* (the magnitude of the TD error) *v.s.* absolute value of EVB (left), EIV (middle) and PIV (right) in DQN. **d-f.** Theoretical upper bound and **g-i.** lower bound *v.s.* absolute value of EVB, EIV and PIV in soft DQN. The red line is the identity line.

Mnih et al. (2015). For each game, the network is trained on a single GPU for 40M frames, or approximately 2 days.

Figure 4 shows that soft DQN prioritized by *surprise* or the theoretical upper bound significantly outperforms uniform replay in most of the games. On average, soft DQN with PER or VER outperform vanilla soft DQN by 11.8% or 18.0% respectively. Moreover, VER shows higher convergence speed and outperforms PER in most of the games (8.47% on average), which suggest that a tighter upper bound on value metrics improves the performance of experience replay. These results suggest that the theoretical upper bound can serve as an appropriate priority for experience replay in soft Q-learning.

## 6 DISCUSSION

In this work, we formulate a framework to study relationship between the importance of experience and *surprise* (the magnitude of the TD error). To quantify the importance of experience, we derive three value metrics of experience: expected value of backup, policy evaluation value, and policy improvement value. For Q-learning, we theoretically show these value metrics are upper bounded by *surprise*. Our claims are supported by the experiments of tabular Q-learning and DQN. Thus, *surprise* implicitly tracks the value of the experience, which leads to high sample efficiency of PER. Furthermore, we extend our framework to maximum-entropy RL, by showing that these value

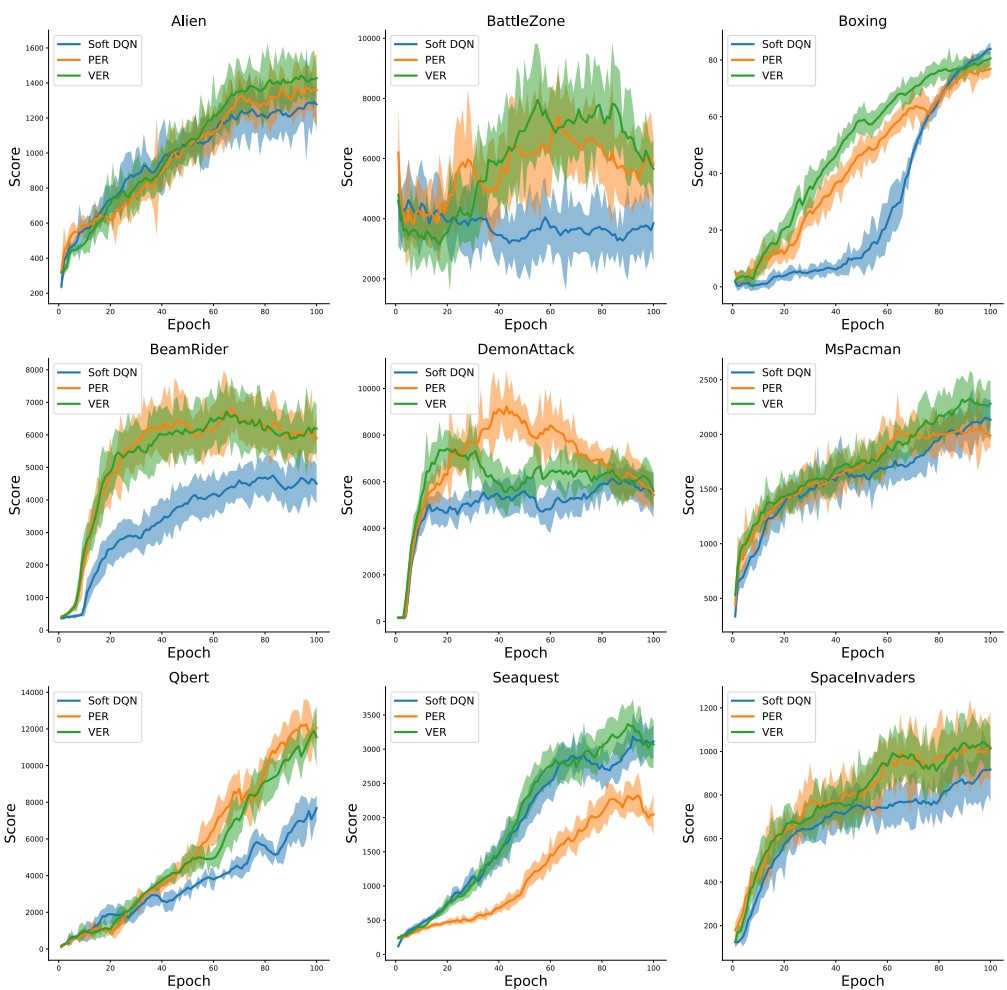

Figure 4: Learning curve of soft DQN (blue lines), and soft DQN with prioritized experience replay in term of soft TD error (PER, orange lines) and the theoretical upper bound of value metrics of experience (VER, green lines) on nine Atari games.

metrics are lower and upper bounded by the product of a policy term and *surprise*. The results in soft Q-learning and soft DQN supports our theory. Moreover, we employ the upper bound as the priority for experience relay, termed as VER, in soft DQN. And we empirically show that VER outperforms PER and significantly improves the sample efficiency of soft DQN.

By linking *surprise* and value of experience, two important quantities in learning, our study has the following implications. First, from a machine learning perspective, our study provide a framework to derive appropriate priorities of experience for different algorithms, with possible extension to batch RL (Fu et al., 2020) and sequence experience replay Brittain et al. (2019). Second, for neuroscience, our work provides insight on how brain might encode the importance of experience. Since *surprise* biologically corresponds to the reward prediction-error signal in the dopaminergic system (Schultz et al., 1997; Glimcher, 2011) and implicitly tracks the value of the experience, the brain may account on it to differentiate important experiences.

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

## A  APPENDIX

### A.1  PROOF OF THEOREM 3.1

In this section, we derive upper bounds of value metrics of experience in Q-learning. The absolute EVB can be written as

$$
\begin{aligned}
|\text{EVB}(e_k)| &= |\max_a Q_{\text{new}}(s_k, a) - \max_a Q_{\text{old}}(s_k, a)| \\
&\leq \max_a |Q_{\text{new}}(s_k, a) - Q_{\text{old}}(s_k, a)| \\
&\leq \alpha |\text{TD}(s_k, a_k, r_k, s'_k)|,
\end{aligned}
\tag{9}
$$

where the second line is from the contraction of max operator.

The absolute PIV can be written as

$$
\begin{aligned}
|\text{PIV}(e_k)| &= |\max_a Q_{\text{new}}(s_k, a) - Q_{\text{new}}(s_k, \arg\max_a Q_{\text{old}}(s_k, a))| \\
&= \max_a Q_{\text{new}}(s_k, a) - Q_{\text{new}}(s_k, \arg\max_a Q_{\text{old}}(s_k, a)) \\
&= \max_a Q_{\text{new}}(s_k, a) - \max_a Q_{\text{old}}(s_k, a) - \mathbf{1}_{a_{\text{old}}=a_k} \alpha \text{TD}(s_k, a_k, r_k, s'_k)
\end{aligned}
\tag{10}
$$

where the second line is from that the change in Q-function following greedy policy improvement is greater or equal to 0, and the third line is from the update of Q-function. For $\text{TD}(s_k, a_k, r_k, s'_k) \geq 0$, we have

$$
0 \leq \max_a Q_{\text{new}}(s_k, a) - \max_a Q_{\text{old}}(s_k, a) \leq \alpha \text{TD}(s_k, a_k, r_k, s'_k).
$$

And for $\text{TD}(s_k, a_k, r_k, s'_k) \leq 0$, we have

$$\max_a Q_{\text{new}}(s_k, a) - \max_a Q_{\text{old}}(s_k, a) \leq 0.$$

Bring above inequalities to 10, we have

$$|\text{PIV}(e_k)| \leq \alpha |\text{TD}(s_k, a_k, r_k, s'_k)| \tag{11}$$

Similarly, the absolute EIV can be written as follows,

$$
\begin{aligned}
|\text{EIV}(e_k)| &= |Q_{\text{new}}(s_k, a_{\text{old}}) - Q_{\text{old}}(s_k, a_{\text{old}})| \\
&= \mathbf{1}_{s=s_k, a_{\text{old}}=a_k} \alpha |\text{TD}(s_k, a_k, r_k, s'_k)| \\
&\leq \alpha |\text{TD}(s_k, a_k, r_k, s'_k)|
\end{aligned}
\tag{12}
$$

For equations (9) and (11), the equality is reached if the experienced action is the same as the best action before and after the update. For (12), the equality is met if the experienced action is the best action before update.

## A.2 PROOF OF THEOREM 4.1

In this section, we derive upper bounds of value metrics of experience in soft Q-learning. For soft-Q learning, $|\text{EVB}^{\text{soft}}|$ can be written as

$$
\begin{aligned}
|\text{EVB}^{\text{soft}}(e_k)| &= |\beta \log \sum_a \exp(\frac{1}{\beta} Q_{\text{new}}^{\text{soft}}(s_k, a)) - \beta \log \sum_a \exp(\frac{1}{\beta} Q_{\text{old}}^{\text{soft}}(s_k, a))| \\
&= |\beta \log \sum_a \exp\left(\frac{1}{\beta}(Q_{\text{old}}^{\text{soft}}(s_k, a) + \mathbf{1}_{a=a_k}\text{TD}^{\text{soft}})\right) - \beta \log \sum_a \exp \frac{1}{\beta} Q_{\text{old}}^{\text{soft}}(s_k, a)|.
\end{aligned}
$$

Let us define the LogSumExp function $F(\vec{x}) = \beta \log \sum_i \exp\left(\frac{x_i}{\beta}\right)$. The LogSumExp function $F(\vec{x})$ is convex, and is strictly and monotonically increasing everywhere in its domain (El Ghaoui, 2018). The partial derivative of $F(\vec{x})$ is a softmax function

$$\frac{\partial F(\vec{x})}{\partial x_i} = \text{softmax}_i(\frac{1}{\beta}\vec{x}) \geq 0,$$

which takes the same form as the policy of soft Q-learning. For $\epsilon < 0$, we have:

$$\epsilon \frac{\partial F(x_1, ..., x_i, ...)}{\partial x_i} \leq F(x_1, ..., x_i + \epsilon, ...) - F(x_1, ..., x_i, ...) \leq 0.$$

Similarly, for $\epsilon \geq 0$, we have,

$$0 \leq F(x_1, ..., x_i + \epsilon, ...) - F(x_1, ..., x_i, ...) \leq \epsilon \frac{\partial F(x_1, ..., x_i + \epsilon, ...)}{\partial x_i}.$$

By substituting $x_i$ by $Q_{\text{old}}^{\text{soft}}(s_k, a_k)$ and $\epsilon$ by $\text{TD}^{\text{soft}}$, and rewriting partial derivative of $F(\vec{x})$ into policy form, we have following inequalities. For $\text{TD}^{\text{soft}} \leq 0$

$$\pi_{\text{old}}(a_k|s_k)\text{TD}^{\text{soft}} \leq \beta \log \sum_a \exp(\frac{1}{\beta} Q_{\text{new}}^{\text{soft}}(s_k, a)) - \beta \log \sum_a \exp(\frac{1}{\beta} Q_{\text{old}}^{\text{soft}}(s_k, a)) \leq 0$$

Similarly, for $\text{TD}^{\text{soft}} > 0$, we have :

$$0 \leq \beta \log \sum_a \exp(\frac{1}{\beta} Q_{\text{new}}^{\text{soft}}(s_k, a)) - \beta \log \sum_a \exp(\frac{1}{\beta} Q_{\text{old}}^{\text{soft}}(s_k, a)) \leq \pi_{\text{new}}(a_k|s_k)\text{TD}^{\text{soft}}$$

Thus, we have the upper bounds of $|\text{EVB}^{\text{soft}}|$,

$$\left|\text{EVB}^{\text{soft}}(e_k)\right| \leq \max\{\pi_{\text{old}}(a_k|s_k), \pi_{\text{new}}(a_k|s_k)\} * \left|\text{TD}^{\text{soft}}\right|.$$

For $|\text{PIV}^{\text{soft}}|$, we have,

$$
\begin{aligned}
|\text{PIV}^{\text{soft}}(e_k)| = |\sum_a \pi_{\text{new}}(a|s)\{Q_{\text{new}}^{\text{soft}}(s_k, a) - \beta \log(\pi_{\text{new}}(a|s))\} \\
- \sum_a \pi_{\text{old}}(a|s)\{Q_{\text{new}}^{\text{soft}}(s_k, a) - \beta \log(\pi_{\text{old}}(a|s))\}| \\
= \sum_a \pi_{\text{new}}(a|s)\{Q_{\text{new}}^{\text{soft}}(s_k, a) - \beta \log(\pi_{\text{new}}(a|s))\} \\
- \sum_a \pi_{\text{old}}(a|s)\{Q_{\text{old}}^{\text{soft}}(s_k, a) - \beta \log(\pi_{\text{old}}(a|s))\} - \pi_{\text{old}}(a_k|s)\text{TD}^{\text{soft}} \\
= \beta \log \sum_a \exp \frac{Q_{\text{new}}^{\text{soft}}(s_k, a)}{\beta} - \beta \log \sum_a \exp \frac{Q_{\text{old}}^{\text{soft}}(s_k, a)}{\beta} - \pi_{\text{old}}(a_k|s)\text{TD}^{\text{soft}},
\end{aligned}
$$

where the second line is because the policy improvement value is always greater than or equal to 0, and the third line is by reordering the equation.

For $\text{TD}^{\text{soft}} > 0$, we have:

$$
0 \le \beta \log \sum_a \exp \frac{Q_{\text{new}}^{\text{soft}}(s_k, a)}{\beta} - \beta \log \sum_a \exp \frac{Q_{\text{old}}^{\text{soft}}(s_k, a)}{\beta} - \pi_{\text{old}}(a_k|s)\text{TD}^{\text{soft}} \le \pi_{\text{new}}(a_k|s_k)\text{TD}^{\text{soft}}
$$

For $\text{TD}^{\text{soft}} \le 0$, we have:

$$
0 \le \beta \log \sum_a \exp \frac{Q_{\text{new}}^{\text{soft}}(s_k, a)}{\beta} - \beta \log \sum_a \exp \frac{Q_{\text{old}}^{\text{soft}}(s_k, a)}{\beta} - \pi_{\text{old}}(a_k|s)\text{TD}^{\text{soft}} \le \pi_{\text{old}}(a_k|s_k)\text{TD}^{\text{soft}}
$$

Thus, we have the upper bounds of $\left|\text{PIV}^{\text{soft}}\right|$:

$$
\left|\text{PIV}^{\text{soft}}(e_k)\right| \le \max\{\pi_{\text{old}}(a_k|s_k), \pi_{\text{new}}(a_k|s_k)\} * \left|\text{TD}^{\text{soft}}\right|
$$

Also, for $|\text{EIV}^{\text{soft}}|$, we have:

$$
\begin{aligned}
|\text{EIV}^{\text{soft}}(e_k)| &= |\sum_a \pi_{\text{old}}(a|s)[Q_{\text{new}}^{\text{soft}}(s_k, a) - Q_{\text{old}}^{\text{soft}}(s_k, a)]| \\
&= \pi_{\text{old}}(a|s) * \left|\text{TD}^{\text{soft}}\right| \\
&\le \max\{\pi_{\text{old}}(a_k|s_k), \pi_{\text{new}}(a_k|s_k)\} * \left|\text{TD}^{\text{soft}}\right|
\end{aligned}
$$

There is no lower bound of the similar form for $|\text{PIV}^{\text{soft}}|$. $\qquad\square$

## A.3 PROOF OF THEOREM 4.2

In this section, we derive lower bounds of value metrics of experience in soft Q-learning. Similar as deriving upper bounds in Appendix A.2, we derive the lower bounds for $|\text{EVB}|$ using the the LogSumExp function $F(\vec{x}) = \beta \log \sum_i \exp(\frac{x_i}{\beta})$. For $\epsilon < 0$, we have:

$$
F(x_1, ..., x_i + \epsilon, ...) - F(x_1, ..., x_i, ...) \le \epsilon \frac{\partial F(x_1, ..., x_i + \epsilon, ...)}{\partial x_i} \le 0.
$$

Similarly, for $\epsilon \ge 0$, we have,

$$
F(x_1, ..., x_i + \epsilon, ...) - F(x_1, ..., x_i, ...) \ge \epsilon \frac{\partial F(x_1, ..., x_i, ...)}{\partial x_i} \ge 0.
$$

By substituting $x_i$ by $Q_{\text{old}}^{\text{soft}}(s_k, a_k)$ and $\epsilon$ by $\text{TD}^{\text{soft}}$, and rewriting partial derivative of $F(\vec{x})$ into policy form, we have following inequalities. For $\text{TD}^{\text{soft}} \le 0$

$$
\beta \log \sum_a \exp(\frac{1}{\beta} Q_{\text{new}}^{\text{soft}}(s_k, a)) - \beta \log \sum_a \exp(\frac{1}{\beta} Q_{\text{old}}^{\text{soft}}(s_k, a)) \le \pi_{\text{new}}(a_k|s_k)\text{TD}^{\text{soft}} \le 0
$$

Similarly, for $\text{TD}^{\text{soft}} > 0$, we have :

$$\beta \log \sum_a \exp(\frac{1}{\beta} Q_{\text{new}}^{\text{soft}}(s_k, a)) - \beta \log \sum_a \exp(\frac{1}{\beta} Q_{\text{old}}^{\text{soft}}(s_k, a)) \geq \pi_{\text{old}}(a_k|s_k)\text{TD}^{\text{soft}} \geq 0$$

Thus, we have the lower bounds of $|\text{EVB}^{\text{soft}}|$,

$$\left|\text{EVB}^{\text{soft}}(e_k)\right| \geq \min\{\pi_{\text{old}}(a_k|s_k), \pi_{\text{new}}(a_k|s_k)\} * \left|\text{TD}^{\text{soft}}\right|.$$

For $|\text{EIV}^{\text{soft}}|$, we have:

$$|\text{EIV}^{\text{soft}}(e_k)| = |\sum_a \pi_{\text{old}}(a|s)[Q_{\pi_{\text{new}}}^{\text{soft}}(s_k, a) - Q_{\pi_{\text{old}}}^{\text{soft}}(s_k, a)]|$$

$$= \pi_{\text{old}}(a|s) * \left|\text{TD}^{\text{soft}}\right|$$

$$\geq \min\{\pi_{\text{old}}(a_k|s_k), \pi_{\text{new}}(a_k|s_k)\} * \left|\text{TD}^{\text{soft}}\right|$$

$\square$

## A.4 EXTENSION TO FUNCTION APPROXIMATION METHOD

In function approximation methods, we learn a parameterized Q-function $Q(s, a; \theta_t)$. The parameter is updated on experience $e_k$ through gradient-based method,

$$\theta_{t+1} = \theta_t + \alpha(Q_{\text{target}}(s_k, a_k) - Q(s_k, a_k; \theta_t))\nabla_{\theta_t} Q(s_k, a_k; \theta_t)),$$

where $\alpha$ is the learning rate and $Q_{\text{target}}$ is the target Q value, defined as

$$Q_{\text{target}}(s_k, a_k) = r_k + \gamma \max_{a'} Q(s_k', a'; \theta_t).$$

And the TD-error is defined as:

$$\text{TD} = Q_{\text{target}}(s_k, a_k) - Q(s_k, a_k; \theta_t).$$

As $\alpha$ in function approximation Q-learning is usually very small, for each update, the parameterized function moves to its target value only by a small amount.

Our framework can be extended to function approximation method by slightly modifying the definition of the value metrics of experience: replacing the Q-function after the update ($Q(s, a; \theta_{t+1})$) by the target Q value ($Q_{\text{target}}(s, a)$) in the value metrics of experience (1-5 and 6-8). The intuition behind this modification is simple: the value is defined by the cause of the update (target Q-value), but not the result of the update through gradient-based update. With this modification our theory is applicable to all function approximation methods, regardless the specific forms of the function approximator (linear function or neural networks). For Q-learning, the value metrics can be written as:

$$\text{EVB}(e_k) = \max_a Q_{\text{target}}(s_k, a) - \max_a Q(s_k, a; \theta_t)$$

$$\text{PIV}(e_k) = \max_a Q_{\text{target}}(s_k, a) - Q_{\text{target}}(s_k, a_{\text{old}})$$

$$\text{EIV}(e_k) = Q_{\text{target}}(s_k, a_{\text{old}}) - Q(s_k, a_{\text{old}}; \theta_t).$$

And for soft Q-learning, the value metrics can be written as:

$$\text{EVB}^{\text{soft}}(e_k) = \beta \log \sum_a \exp(\frac{1}{\beta} Q_{\text{target}}^{\text{soft}}(s_k, a)) - \beta \log \sum_a \exp(\frac{1}{\beta} Q^{\text{soft}}(s_k, a; \theta_t))$$

$$\text{PIV}^{\text{soft}}(e_k) = \beta \log \sum_a \exp \frac{Q_{\text{target}}^{\text{soft}}(s_k, a)}{\beta} - \beta \log \sum_a \exp \frac{Q^{\text{soft}}(s_k, a; \theta_t)}{\beta} - \pi_{\text{old}}(a_k|s)\text{TD}^{\text{soft}}$$

$$\text{EIV}^{\text{soft}}(e_k) = \sum_a \pi_{\text{old}}(a|s)[Q_{\text{target}}^{\text{soft}}(s_k, a) - Q^{\text{soft}}(s_k, a; \theta_t)].$$

After the modifications, the value metrics of experience have similar form as the tabular case, and all Theorems derived in the tabular case can be applied to function approximation methods.

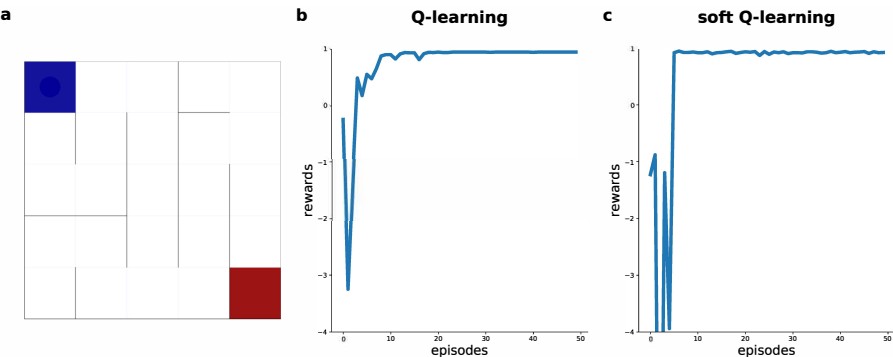

Figure 5: Maze environment and learning curves.

## A.5 EXPERIMENTAL DETAILS

### A.5.1 MAZE

For the maze experiments in section 5.1, we use a maze environment of a $5 \times 5$ square with walls, as depicted in Figure 5a. The agent needs to reach the goal zone in the bottom-right corner. At each time step, the agent can choose to move one square in any of the four directions (north, south, east, west). If the move is blocked by a wall or the border of the maze, the agent stays in place. Every time step, the agent gets a reward of $-0.004$ or $1$ if it enters the goal zone and the episode ends. The discount factor is $0.99$ throughout the experiments. For these experiments, we use a tabular setting for Q-learning and soft Q-learning according to section 2.1 and 4.1. For Q-learning, the behavior policy is $\epsilon$-greedy, where $\epsilon$ decays exponentially from 1 to 0.001 during training. And we set learning step size $\alpha = 1$. For soft Q-learning, the temperature parameter $\beta$ is set to 100. Total trial number is 50 for each algorithm. During training, both algorithms successfully solve the maze game, see Figure 5b-c for the learning curves.

### A.5.2 CARTPOLE

For CartPole, the goal is to keep the pole balanced by moving the cart forward and backward for 200 steps. We test our theoretical prediction on DQN and soft-DQN (DQN with soft-update). For DQN, we implement the model according to Mnih et al. (2015), where we replace the original Q-network with a two-layer MLP, with 256 Relu neurons for each layer. The $\epsilon$ in $\epsilon$-greedy policy decays exponentially from 1 to 0.01 for the first $10,000$ steps, and remains 0.01 afterwards. For soft-DQN, all settings are the same with DQN, except for two modifications: for policy evaluation, the (soft) Q-network is updated according to the soft TD error; the policy follows maximum-entropy policy, calculated as the softmax of the soft Q values (see section 4.1). The temperature parameter $\beta$ is set to 0.5. For both algorithms, the discount factor is 0.99, the learning rate is 0.005, experience buffer size is 1000, the batch size is 16 and total environment interaction is $50,000$.

### A.5.3 ATARI GAMES

For this set of experiments, we compare the performance of vanilla soft DQN and soft DQN with PER, where we use *surprise* and the theoretical upper bound as priorities (Schaul et al., 2016), respectively denoted as PER and VER (valuable experience replay). We select 9 Atari games for the experiments: Alien, BattleZone, Boxing, BeamRider, DemonAttack, MsPacman, Qbert, Seaquest and SpaceInvaders. The vanilla soft DQN is similar to that described in the above section, but the Q-network the same with Mnih et al. (2015). We implement PER on soft-DQN according to Schaul et al. (2016). For all algorithms, the temperature parameter $\beta$ is 0.05, the discount factor is 0.99, the learning rate is $1e{-}4$, experience buffer size is 1M, the batch size is 32, total environment interaction is $50,000$. For PER or VER, the parameters for importance sampling are $\alpha_{\text{IS}} = 0.4$ and $\beta_{\text{IS}} = 0.6$. For each game, the network is trained on a single GPU for 40M frames, or approximately 2 days.

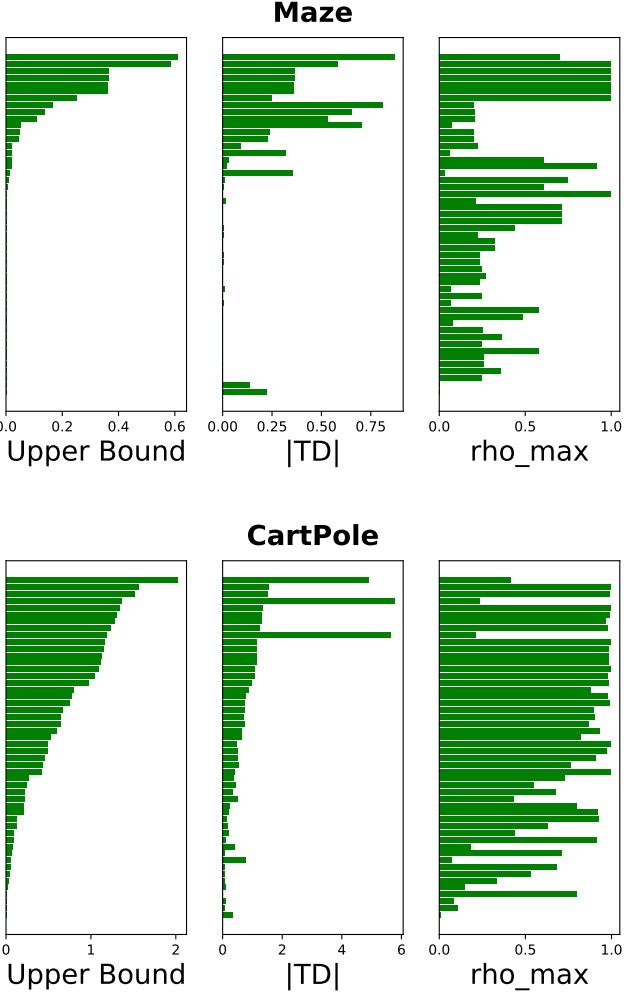

Figure 6: Illustration on the difference between VER and PER in soft Q-learning. VER uses the theoretical upper bound as priority ($\rho_\pi^{\max} * |\text{TD}^{\text{soft}}|$), which balances the TD error and the "on-policyness" of the experience. Depicted are the theoretical upper bound (left), $|\text{TD}|$ (middle), and the policy term (right) of 50 experiences from the replay buffer in the maze (upper panel) and CartPole (lower panel), ordered by the theoretical upper bound.

