# OpenReview forum: "Revisiting Prioritized Experience Replay: A Value Perspective"
_ICLR.cc/2021/Conference — Reject_

### Official Review · AnonReviewer4 · 2020-10-28
**Very interesting work with great potential, but the choice of EVB needs clearer motivation, and experimental is inconclusive**

**Rating:** 4
**Confidence:** 4

**Review:**

Summary: the idea of prioritized experience replay is revisited, but from a new perspective with new theoretical results. Here, the authors propose the expected value of backup (EVB) as a metric to assess the quality of a sample and its potential improvement on the policy and on the value function. The authors decompose this metric into the benefit attributed to the policy and benefit to the value function. The authors have two theorems. The first theorem shows that the surprise (aka the temporal difference error) is an upper bound of the EVB in the Q-learning setting. The second theorem shows that the surprise, multiplied by a constant that depends on the policy, is an upper bound to EVB in the soft RL setting. The authors demonstrate that the proposed tighter bound on the EVB *could* yield improvements in the soft RL setting.

Pros of this work:
- the work tries to tackle an important and still not fully answered question, which is why prioritized replay works well in the DQN setting but not in the soft RL settings, making very nice connections to the existing empirical literature on the topic, and a suggestion of how to improve PER
- the paper is very clearly written and the motivation of addressing PER from a rigorous perspective is clear.
- overall, a rigorous study of the "quality" of samples and experience seems like a very promising direction if the goal is to develop more intelligent agents that can make better use of the information available to them
- the authors try to demonstrate why the approach leads to tighter bounds than standard PER with some visualization (although I point out some issues below)

Cons:
- lack of motivation regarding the choice of EVB
- more thorough experimentation
- lack of evidence suggesting a tight upper bound (e.g. modest improvement in soft RL, maze example does not suggest a tight bound in Q-learning)

Details:
- lack of motivation regarding the choice of EVB: while the EVB seems like an intuitive starting point for investigation, its motivation and a-priori connection to prioritized replay is not fully clear. one of the problems with EVB is that it seems to be a largely myopic measure of quality that is linked to only the current sample, e.g. (s,a,r,s'), while ignoring the effect of the sample on further backups in the rollout. Recent work on the topic [1] (PSER) suggest that a myopic approach can be significantly improved by up-weighting earlier transitions that lead to good transitions in the future. This would seem to suggest that a non-myopic metric could lead to more significant improvements than working with a myopic setting which could be fundamentally flawed. While I do agree that EVB is a good starting point for analysis, I am not convinced the results are fully conclusive, particularly given the need for stronger motivation on EVB and more conclusive experiments (see below) . I would encourage the authors to think about how to better motivate the use of EVB in the papers introduction more clearly if existing literature suggests so (I am somewhat aware of [2], although the authors there do highlight the shortcomings of EVB). Could sequential replay be the result of using an underlying metric that is less myopic than EVB, which seems to shows better promise over PER? If not, why is EVB truly the only right approach for analyzing PER?
- more thorough experimentation: while I concur that the current paper addresses an important theoretical question, and while the result appears trivial to implement (not more difficult than PER), the bound should also be justified experimentally if it is to have significant value in practice. I think one key detail that is missing and could greatly benefit the paper is a more careful analysis and visualization of what the modified tighter bound is actually doing in the soft-Q setting. For example, how does the ranking of experience change when using the "tighter" bound? How often does this bound lead to a revision or re-evaluation of experience v.s. PER? Is there a fundamental reason why PER could not be as effective for soft RL in general that cannot be explained by better myopic estimates of the value of each transition?
- lack of evidence suggesting a tight upper bound - Q: while the authors argue at several points in the paper that the |TD| could be a tight upper bound, this bound is only attained in some special cases. However, this does not really mean that the upper bound is tight for Q-learning, since there could be other upper bounds based on |TD| that incorporate additional value or policy information that could be tighter (same also applies for soft RL). The Figure 1 also seems to suggest that the values are scattered quite randomly and do not attain the upper bound as claimed for Q-learning. This does not suggest that |TD| is tight in any way. Can a gap be proved that effectively provides a lower bound on EVB?
- lack of empirical evidence supporting the tighter bounds - soft RL: The experiments on Atari also suggest that the improvements of VER are quite mild, only improving on PER with statistical confidence on two of the experiments (interestingly, these two experiments also show mild improvement by PSER, whereas PSER shows considerable improvements on other games evaluated here).  This seems to be at odds with the claims that VER is a significant improvement of PER in soft RL. In the games where VER does not improve upon PER significantly, I would encourage the authors to comment on why the results are so similar to PER. I think the current benchmark problems are fairly sufficient and complex, and would not require more evaluation, unless the authors believe this would lead to a different conclusion or if the games selected are not representative of where VER can be beneficial.

Other points:
- while I agree with the authors that when the learning rate is held constant, |TD| is an "upper bound" to EVB in the Q-learning case. However, in practice we often use state-dependent learning rates that can be annealed over time with visitation counts, which can often yield improvements (as long as the usual stochastic convergence conditions are satisfied, of course). In this case, wouldn't the learning rate play a role in the bound?

I think the work is very interesting and addresses a central issue, and I hope that the above comments can be useful to improve the paper. Overall, I think it is necessary to think more carefully about the connection between PER and quantifying the value of an experience (e.g. why EVB? how to reconcile moderate empirical evidence of the new bounds?). I am looking forward to the authors' response on these issues above.

References:
[1] Brittain, Marc, et al. "Prioritized Sequence Experience Replay." arXiv preprint arXiv:1905.12726 (2019).
[2] Mattar, Marcelo G., and Nathaniel D. Daw. "Prioritized memory access explains planning and hippocampal replay." Nature neuroscience 21.11 (2018): 1609-1617.

---

> ### Author Response · Authors · 2020-11-24
> **Responses to Reviewer 4**
>
> Thanks for your interests, comments and suggestions on our work, which are encouraging, critical and helpful to improve our work. We have made several modifications on the manuscripts following your suggestions and concerns. Our responses to the comments are listed below:
>
> > Lack of motivation regarding the choice of EVB
>
> You are right, we did a terrible job on introducing and motivating the choice of EVB. Now we add a new paragraph in the introduction section, and a motivating example (section 2.4) that explains we should use EVB as the correct metric for prioritization. The new paragraph in the introduction section is as follows:
>
> _"We address this problem from an economic perspective, by linking the surprise to the value of experience in RL. The goal of RL agent is to maximize an expected cumulative reward, which is achieved through learning from experiences. For Q-learning, an update on an experience will lead to a more accurate prediction of the action-value or a better policy, which increases the expected cumulative reward the agent may get. We define the value of experience as the increase in the expected cumulative reward resulted by updating on the experience (Mattar et al. 2018), which quantifies the importance of experience from first principles: assuming that the agent is economically rational and has full information about the value of experience, it will choose the experience with most value for update, which will yield the highest utility. As supplements, we derive two more value metrics, which corresponds to the evaluation improvement value and policy improvement value due to update on experience."_
>
> The motivating example tries to compare different prioritization strategies in a “Linear GridWorld” environment, showing that prioritization with EVB is optimal. Please check the updated manuscript for details.
>
> We agree with you that it is beneficial to use a sequence of transitions (experiences) rather than single transition, because using N-step returns (Hessel et al. 2017) and Prioritized  Sequence Experience Replay (PSER, Brittain et al. 2019) may outperform myopic method. In fact, our framework is not exclusive to myopic settings. We can also extend our framework to a sequence of transitions (experiences): first define the appropriate value metrics for the sequence of transitions, and then derive their theoretical bounds. We think it is a promising extension to our current work.
>
> > More thorough experimentation
>
> For Q-learning, we add a new figure to demonstrate the difference between prioritization with surprise (|TD|) and EVB, and explain why EVB is an optimal choice to prioritize experiences. For soft Q-learning, the difference between the "tighter" bound and |TD| used in PER is that the "tighter" bound involves a policy term that quantifies the “on-policyness” of the experience, which is critical for off-line learning (Schlegel et al. 2019). To illustrate how the ranking of experiences changes when using the "tighter" bound, we add a new figure (Figure 6) in Appendix to illustrate how the "tighter" bound balances |TD| and “on-policyness” of experiences.
>
> > Lack of empirical evidence supporting the tighter bounds - soft RL
>
> To show that the bounds is tight in soft Q-learning, we additionally derive the lower bound of EVB and EIV (but not PIV), which have similar form as the upper bound that involves a policy term and surprise (Theorem 4.2). And we verify the lower bounds hold true in practice with Maze and CartPole environments.
>
> There are games where VER does not improve upon PER significantly. We think it is because the |TD| alone is the “softer” bound of the EVB and may work well without correcting for the “on-policyness” in some games.
>
>
> > Extension to state-dependent learning rates that can be annealed over time with visitation counts
>
> From our understanding, state-dependent learning rates that can be annealed over time with visitation counts have similar motivation as the count-based exploration, which encourages learning in the state that is more novel. In tabular case, a more frequently visited state has a learning rate that is lower compared to less frequently visited state, which results a lower value in the associated experience.We think a simple modification will do the trick: when store the experience $(s, a, r, s')$ into the replay buffer, scale the original upper bound (priority) by a constant that annealed over time with visitation counts.
>
>
> *Reference:*
>
> - Hessel M, Modayil J, Van Hasselt H, et al. Rainbow: Combining improvements in deep reinforcement learning. arXiv:1710.02298, 2017.
>
> - Brittain, Marc, et al. Prioritized Sequence Experience Replay. arXiv:1905.12726, 2019.
>
> - Schlegel M, Chung W, Graves D, et al. Importance resampling for off-policy prediction. Advances in Neural Information Processing Systems, 2019.

---

### Official Review · AnonReviewer2 · 2020-10-28

**Rating:** 5
**Confidence:** 4

**Review:**

This work aimed to understand the prioritized experience replay, a widely used technique to improve learning efficiently for RL agents. The authors proposed three different value metrics to quantify the experience, and showed that they are upper bounded by the TD error (up to a constant). The extension to soft Q-learning was also presented. Finally, the authors showed in experiments that derived upper bounds hold in Maze and CartPole. They also demonstrated that a new variant based on the upper bound achieved better performance on a subset of Atari games.

The authors tried to achieve a deep understanding of the prioritized experience replay, which I believe to be an important task. However, after reading through the paper, I am afraid that the question why prioritized experience replay works so well in practice is not well addressed. The authors provided the three metrics, and provided their upper bounds. Unfortunately, they are not sufficient to help the understanding, as the upper bound derived is just the TD error used in the prioritized experience replay. I also do not see enough depth for these theoretical results, and the presentation could be improved as well. The experiments indeed showed some practical benefits, but descriptions are confusing sometimes. My detailed comments and questions are as follows.

1. When defining EVB, PIV, EIV in Eq. (3)- Eq. (5), "s_k" is used. However, in the derivation of Theorem 3.1, why is "s" used? They should be the same if both are from "e_k".
2. The introduction of function approximators in the last paragraph in Section 2 is confusing: How are they used in the following sections? Also, why does the assumption "...as if the parameterized Q-function converges to its target value" hold?
3. For Theorem 3.1, it looks to me that a tighter upper bound according to the derivations should be \alpha |TD|. Why did you omit \alpha?
4. In the derivation for Eq. (7), the authors claimed that "the third line is because the increase in Q-function resulted from greedy
policy improvement will not exceeds the surprise (times the learning step-size)". Could you elaborate more on why this is the case?
5. For VER in Section 5.3, which upper bound does it use? I guess it may come from Theorem 4.1, but need more clarification. Also, what are the exact differences between VER and PER?

---

> ### Author Response · Authors · 2020-11-24
> **Responses to Reviewer 2**
>
> Thanks for your comments and suggestions, which are critical and helpful to improve our work. To improve the understanding of the value metrics of experience, especially the expected value of backup (EVB), we add a paragraph (Introduction Paragraph 3) and a new motivating example (section 2.4) in the updated manuscript. To make theoretical result deeper, we derive theoretical lower bounds for EVB and EIV (but not PIV) in soft Q-learning ($\rho^\text{min}_\pi * | \text{TD}^{\text{soft}}|$), which have similar form as the upper bound that involves a policy term and surprise (Theorem 4.2). And we verify the lower bounds hold true in practice using Maze and CartPole environments. We also made corrections and clarifications regarding your detailed comments and questions:
>
> > When defining EVB, PIV, EIV in Eq. (3)- Eq. (5), "s_k" is used. However, in the derivation of Theorem 3.1, why is "s" used? They should be the same if both are from "e_k".
>
> Yes, they should be the same. We have corrected them in the updated manuscript.
>
> > The introduction of function approximators in the last paragraph in Section 2 is confusing: How are they used in the following sections? Also, why does the assumption "...as if the parameterized Q-function converges to its target value" hold?
>
> Yes, we found this part is confusing. The value metrics of experience are used in the function approximation methods similar as in the tabular case, by substituting the $Q_{new}$ to $Q_{target}$ in equation (3)-(5). "...as if the parameterized Q-function converges to its target value" is not an assumption, they are defined this way so that the theory can be applied directly to function approximation methods. The intuition behind this definition is simple: the value is measure from the cause of the update (target Q-value), but not the result of the update (specific gradient). This makes our theory applicable to all function approximation methods, regardless the form of the function approximator (linear approximator or neural networks). We elaborate this more in the updated manuscript (see Appendix A.4).
>
> > For Theorem 3.1, it looks to me that a tighter upper bound according to the derivations should be \alpha |TD|. Why did you omit \alpha?
>
> We add $\alpha$ in the updated manuscript.
>
>
> > In the derivation for Eq. (7), the authors claimed that "the third line is because the increase in Q-function resulted from greedy policy improvement will not exceeds the surprise (times the learning step-size)". Could you elaborate more on why this is the case?
>
> Yes, we add details for this step in the updated manuscript. Please note that the proof of Theorem 3.1 is placed to Appendix A.1.
>
>
> > For VER in Section 5.3, which upper bound does it use? I guess it may come from Theorem 4.1, but need more clarification. Also, what are the exact differences between VER and PER?
>
> Yes, they are from Theorem 4.1. PER is prioritized by *surprise*, and VER is prioritized by the derived upper bound ($\rho^\text{max}_\pi * | \text{TD}^{\text{soft}}|$). To make it clearer, we add more description in the beginning of the experiment section as well as in Section 5.3.

---

### Official Review · AnonReviewer1 · 2020-10-30
**Not convinced about motivation, method or results**

**Rating:** 3
**Confidence:** 4

**Review:**

This paper tries to interpret PER from the lens of replaying experience that has the most "surprise" and shows how it connects to some notions of value such as the expected value of the backup and policy improvement value and evaluation improvement value and argue. The authors also derive a max-ent version of this and show that this can improve performance on some Atari games (though this is not that convincing).

My score for this paper is based on these points:

 Motivation: I do not see the motivation for introducing these metrics and why that explains PER in the first place. Agreed that PER is a reasonable choice, and it can upper bound the EIB and EVB metrics (i have issues with this too, more on this next), it just seems to me that the paper doesn't make any convincing claim for why this helps us understand why PER works. If the focus of the paper is on understanding PER, then the paper does not do a good job of it. If it is to introduce these prioritization based on these metrics -- and the paper focuses entirely on them -- then I then have several concerns next.

- Definition of the value metrics: The cited definition of these metrics requires using the true Q-function or the true value function of the resulting policy. If we end up approximating this using the learned model, what is the guarantee that these metrics are indeed useful? Also theorem 1 should be restated to say that they care about the "empirical" EIB and EVB, that is computed using the learned Q-function, else it doesn't make sense to me. Moreover, if the TD error is a bound (which I think isn't with neural networks as I discuss in the next point) on the empirical EVB, can't I just drastically overestimate Q-values and get a larger empirical EVB value to be super high and prioritize on those examples? Why is that good? Why won't that promote overestimation?

- Why is the update on the Q-value assumed to be tabular if the experiments are with a deep network on Atari? In a non-tabular setting Theorem 1 does not hold so either that should be rederived for the case of DQN or the experiments should be adjusted to do it on tabular settings.   In any case, now it is not clear to me why the method works with DQN, since the update in this setting isn't equal to $Q(s, a) \leftarrow Q(s, a) + \alpha TD(s, a)$. In general, the solution isn't known with neural networks, so the upper bound story doesn't hold there. With the NTK (Jacot et al.) assumption, I can obtain a somewhat similar update but pre-conditioned with the kernel Gram matrix (see Achiam et al. Towards characterizing divergence in deep Q-learning). However, Theorem 1 doesn't hold anymore now. So, it is unclear why the method works.

- Even if I were to look at the experiments only, the results are not that impressive. The method is generally close to PER, and maybe a little better, but no comparison is made on a more efficient method such as Rainbow, and there are only 9 Atari games, which is too little. So, that is not super convincing yet.

I would suggest the authors make some of the changes above.

---

> ### Author Response · Authors · 2020-11-24
> **Responses to Reviewer 1**
>
> Thanks for your valuable feedback. We think that because the motivation of the EVB is poorly written, you may have some misunderstanding on several aspects of our work. To make the motivation behind EVB clearer, we add a new paragraph in the introduction section, and a motivating example (Section 2.4) that explains why EVB is the optimal metric for prioritization. Intuitively, the goal of RL is to maximize the expected cumulative reward, and EVB directly quantifies the utility of experiences by how they will increase the expected cumulative reward. By drawing a relationship between EVB and surprise (the magnitude of TD error), we can better understand why PER works so well. Further, based the understanding of EVB and surprise, we can derive the prioritization for soft Q-learning. Our responses to the comments are listed below:
>
> > Definition of the value metrics compare to Matter & Daw 2018
>
> We did not clearly state the difference between EVBs in our work and Matter & Daw 2018, which made it easy to misunderstand. Yes, the cited definition of these metrics requires using the true Q-function or the true value function of the resulting policy. We define them in the same way with equation 1 & 2. The difference between Matter & Daw 2018 and our work is how we “empirically” estimate the EVB, as the true value is never known. In their case, EVB is estimated in the Dyna framework (Sutton 1990), a model-based RL algorithm. As a result, the estimation of the EVB is the product of a need term (successor representation) and a gain term (eq.13 in Methods of their paper). During learning, besides learning Q-function, they need to estimate successor representation by learning the state transition probability (which is hard for large state-space problems). In our case, the EVB is derived in Q-learning, a model-free algorithm, so that EVB is estimated by the learned Q-functions. To emphasize, we clarify the difference between the two methods in Section 2.3 of the updated manuscript We agree that overestimation of the Q-values as well as the TD errors impairs the prioritization, just as they impair the performance and stability of Q-learning. We think this can be mitigated by DDQN or other methods that mitigate the overestimation of Q-functions.
>
> > Why is the update on the Q-value assumed to be tabular if the experiments are with a deep network on Atari?
>
> We modified the extension to function approximation method to make it better to understand (Appendix A.4 of the updated manuscript). The value metrics of experience are used in the function approximation methods similar as in the tabular case, by substituting the $Q_{new}$ to $Q_{target}$ in equation (3)-(5). They are defined this way so that the theory can be applied directly to function approximation methods. The intuition behind this modification is simple: the value is measureed from the cause of the update (target Q-value), but not the result of the update (specific gradient). This makes our theory practically applicable to all function approximation methods, regardless the form of the function approximator (linear approximator or neural networks). We elaborate this more in the updated manuscript (see Appendix A.4).
>
>
> > The results are not that impressive
>
> There are games where VER does not improve upon PER significantly. We think it is because the |TD| alone is the “softer” bound of the EVB and may work well without correcting for the “on-policyness” in some games. Rainbow is built on DQN but not soft-DQN, and may not be appropriate for our study.
>
> Atari benchmark is typically used in two ways: using the entire 57 games (Schaul et al 2016) or picking a representative set of the games (Schulman et al 2017). We choose the latter because it is more efficient and less computing-power consuming, and our work focuses on theoretical analysis.
>
>
> *Reference:*
>
> - Mattar M G, Daw N D. Prioritized memory access explains planning and hippocampal replay. Nature neuroscience, 2018, 21(11): 1609-1617.
>
> - Sutton R S. Integrated architectures for learning, planning, and reacting based on approximating dynamic programming. Machine learning proceedings 1990. Morgan Kaufmann, 1990: 216-224.
>
> - Schaul T, Quan J, Antonoglou I, et al. Prioritized experience replay. arXiv:1511.05952, 2015.
>
> - Schulman J, Chen X, Abbeel P. Equivalence between policy gradients and soft q-learning. arXiv:1704.06440, 2017.

---

### Official Review · AnonReviewer3 · 2020-10-31
**Dissecting the validity of using TD-error for prioritized experience replay**

**Rating:** 6
**Confidence:** 3

**Review:**

Summary: The authors of this paper make a connection between the TD-error from a single unit of experience and various metrics of improvement for agents trained with prioritized experience replay and Q-learning or soft Q-learning. They show that priorities based on TD-error are indeed sensible, and show that a small adjustment to TD-error-as-priority for soft Q-learning agents is both theoretically sound and can yield improved performance.

The set-up and motivation of the paper is relatively clear and well-explained. One reason I like the paper is that the process is quite straightforward: 1) Strive to better understand a commonly-used algorithm, 2) Derive theory with reasonably good intuition behind it, 3) show that it empirically holds true on simple environments, and 4) the better understood result also yields modest improvements on a test suite.

There are a few areas I would consider rewriting or rewording for added clarity.

For instance, the thesis of the paper relies on a sensible definition of "value of experience", and this isn't made concrete until later, though perhaps this is hard to do in the introduction. As a small nit, I think the extra use of the word surprise was at first a bit unclear, especially as it aliases TD-error (unless I missed something).

I think an extra paragraph about why we should use surprise as the correct metric for prioritization would be helpful. Notably, that when we know it upper bounds the three metrics, we want to continue prioritizing sampling experiences when training our agent, because this will yield faster learning, since we will have larger improvements to our agent.

However, this dovetails into an additional few questions that could be followed up here: Is there a correct temperature with which we sample experiences from our prioritized replay that is ideal? Should we not be sampling experiences, and simply sorting the experiences by priority and training our agent on experiences in descending order of priority? Why shouldn't we do this? How much worse is this than using uniform sampling? I understand that some of these questions are hard to answer with limited compute, but some mention of this would give the paper more depth.

Additionally, there are a number of small issues in the writing, notably towards the end. A careful read of the paper for proper grammar, making sure the right propositions are used and that there are no missing words in sentences would help the flow and readability of the paper greatly.

While the overall result is a nice one, I believe the paper has somewhat limited scope. It doesn't fundamentally change how we should approach training our agents with a replay buffer. In fact, I suspect that TD-error was used by the original authors because they knew of a link like this, or had strong suspicions of it. I think what this paper should do is explore or at least pose a gamut of interesting follow-up questions about the role of replay and how best to use it. Is it possible that there is instead a lower bound we can derive; can we learn a sampling/prioritization scheme by gradient descent that somehow does better? Are there prioritizations completely disjoint from TD-error that we should consider using? In addition, while limited resources might make more empirical investigations challenging, it's also worth understanding how other commonly used algorithmic mechanisms in deep RL interact with prioritization, such as stepping environments in batches, or the preprocessing done to observations, or things like reward or advantage clipping.

I think that if the paper showed more evidence of zooming out and thinking deeply about the core problem, this would be an excellent paper.

---

> ### Author Response · Authors · 2020-11-24
> **Responses to Reviewer 3**
>
> Thanks for your comments, suggestions and “like” on our work, which are critical, helpful and encouraging. We have made several modifications on the manuscripts following your suggestions and concerns. Our responses to the comments are listed below:
>
> >The thesis of the paper relies on a sensible definition of "value of experience", and this isn't made concrete until later.
>
> You are right, we did a terrible job on introducing and motivating the "value of experience". Now we add a new paragraph in the introduction section, and a motivating example (Section 2.4) that explains why EVB is the optimal metric for prioritization. The new paragraph in the introduction section is as follows:
>
> _“We address this problem from an economic perspective, by linking the surprise to the value of experience in RL. The goal of RL agent is to maximize an expected cumulative reward, which is achieved through learning from experiences. For Q-learning, an update on an experience will lead to a more accurate prediction of the action-value or a better policy, which increases the expected cumulative reward the agent may get. We define the value of experience as the increase in the expected cumulative reward resulted by updating on the experience (Mattar et. al. 2018), which quantifies the importance of experience from first principles: assuming that the agent is economically rational and has full information about the value of experience, it will choose the experience with most value for update, which will yield the highest utility. As supplements, we derive two more value metrics, which corresponds to the evaluation improvement value and policy improvement value due to update on experience."_
>
> The motivating example tries to compare different prioritization strategies in a “Linear GridWorld” environment, showing that prioritization with EVB is optimal. Please check the updated manuscript for details.
>
> > Is it possible that there is instead a lower bound we can derive?
>
> Actually, it is possible. In the updated manuscript, we derived the lower bounds for EVB and EIV (but not PIV) in soft Q-learning ($\rho^\text{min}_\pi * | \text{TD}^{\text{soft}}|$), which have similar form as the upper bound that involves a policy term and surprise. We also show the theoretical lower bound holds true in experiments. Please check the Theorem 4.2 and experiments in the updated manuscript for details.
>
> > Should we not be sampling experiences, and simply sorting the experiences by priority and training our agent on experiences in descending order of priority?
>
> In the tabular case, it is optimal to sort the experiences by priority and train our agent on experiences in descending order of priority (see the motivating example). But this greedy prioritization has some disadvantages, especially for function approximation methods (Schaul et al. 2016). The main issue is that greedy prioritization may focuses on a small subset of important experience, which introduces overfitting and breaks the *i.i.d.* assumption of stochastic gradient-based algorithms. Our preliminary simulation shows that DQN with greedy prioritization does not converge.
>
> > Can we learn a sampling/prioritization scheme by gradient descent that somehow does better?
>
> This is an interesting question, one work in this direction was published in IJCAI-19 (Zha et al. 2019), where they propose the experience replay as an optimization problem and train the replay policy by gradient descent, which outperforms PER in several tasks. Interestingly, the replay policy is updated based on the improvement of cumulative reward, which is related to the value of experience defined in our work.
>
> > Are there prioritizations completely disjoint from TD-error that we should consider using?
>
> Yes, there are. For example, the degree of “on-policyness” (such as the policy related term in the theoretical upper bounds in soft q-learning) of the experience can be serve as a priority, which may work similar as the importance resampling (Schlegel et al. 2019). We are currently evaluating the performance of this prioritization scheme. While the simulation may not be finished before the rebuttal deadline, we will add the results in the final manuscript. Other priorities may involve: (a) state-space coverage; (b) correlation between transitions; (c) cardinality of distribution support (Fedus et al. 2020).
>
> *References:*
>
> - Mattar M G, Daw N D. Prioritized memory access explains planning and hippocampal replay. Nature neuroscience, 2018, 21(11): 1609-1617.
>
> - Schlegel M, Chung W, Graves D, et al. Importance resampling for off-policy prediction. Advances in Neural Information Processing Systems, 2019.
>
> - Fedus W, Ramachandran P, Agarwal R, et al. Revisiting fundamentals of experience replay. arXiv:2007.06700, 2020.

---

### Author Response · Authors · 2020-11-24
**To all the reviewers**

We thank all the reviewers for the insightful  comments. Based on them, we have substantially improved the manuscript  by *(1)*  better motivation: a new paragraph in introduction as well as a motivating example  (Section 2.4), *(2)*  deriving  the lower bounds for EVB and EIV for soft Q-learning  (Theorem 4.2) and verifying  them  by  additional  experiments, *(3)*  rewriting  the  extension to function approximation (Appendix A.4),  and *(4)*  adding a figure (Figure 6) to better illustrate how the  upper bounds balance the TD-error and "on-policyness." Please refer to the revision for details.

---

### Decision · Program_Chairs · 2021-01-07
**Final Decision**

**Decision:**

Reject

**Comment:**

This paper is certainly on the way to be a solid contribution: it's an interesting research question, and we need more understanding papers (rather than yet another algorithmic trick paper).

The reviewers thought the paper was not yet ready. The reviewers suggested: (1) more motivation of why the proposed metrics were of interest, (2) clearer discussion and evidence of how the analysis better articulates the performance of PER, (3) missing empirical details like methodology for setting hyper-parameters, why these 9 Atari games, undefined errorbars, unspecified number of runs, and (4) conclusions not supported by evidence in Atari: with missing experiment details, likely too few runs, and overlapping errorbars in most games few scientific conclusions can be drawn.

The work might be strengthen by developing the first part of the paper (and focussing on the reviewer's suggestions) and deemphasizing the novel algorithmic contribution part.